# Maternal Magnolol Supplementation during Pregnancy and Lactation Promotes Antioxidant Capacity, Improves Gut Health, and Alters Gut Microbiota and Metabolites of Weanling Piglets

**DOI:** 10.3390/metabo13070797

**Published:** 2023-06-27

**Authors:** Qiwen Fan, Encun Du, Fang Chen, Wenjing Tao, Na Zhao, Shaowen Huang, Wanzheng Guo, Jing Huang, Jintao Wei

**Affiliations:** 1Key Laboratory of Animal Embryo Engineering and Molecular Breeding of Hubei Province, Wuhan 430064, China; qwfan11@hbaas.com (Q.F.); taowenjing1127@hbaas.com (W.T.);; 2Key Laboratory of Prevention and Control Agents for Animal Bacteriosis (Ministry of Agriculture), Wuhan 430064, China

**Keywords:** magnolol, weanling piglets, gut health, microbiome, metabolome

## Abstract

Maternal nutrition exerts a profound effect on the postnatal performance of offspring, especially during the weaning period. The multifunctional bioactive component magnolol (MAG) has shown promise as a dietary supplement. This study aimed to explore the effects of maternal MAG supplementation on the antioxidant capacity, gut health, gut microbiome, and metabolome composition of weanling piglets. Fifty pregnant sows were randomly divided into two equally sized groups, the control group and the group supplemented with 100 g/t MAG during the gestation and lactation periods, and 7 days postweaning, the pups were euthanized. The microbiome and metabolome features of weanling piglet colons were compared. Our results revealed that maternal MAG supplementation modified the serum redox status of weanling piglets by decreasing malondialdehyde concentration and increasing superoxide dismutase activity and total antioxidant capacity. Moreover, the decreased indicators of diarrhea were accompanied by improved gut barrier function, in which serum diamine oxidase concentration was decreased, and expressions of zona occludens-1, claudin-1, and intestinal alkaline phosphatase were increased in the colon of weanling piglets from sows supplemented with MAG. Further analysis of the gut microbiota indicated that maternal MAG supplementation significantly increased the relative abundance of beneficial bacteria in the colon of weanling piglets, including *Faecalibacterium prausnitzii* and *Oscillospira*. Metabolome analysis identified 540 differential metabolites in the colon of piglets from MAG-fed dams, of which glycerophospholipid classes were highly correlated with progeny gut health and key beneficial bacteria. Our findings indicated that maternal MAG supplementation can improve the oxidative status and gut health of weanling piglets, possibly due to alterations in the gut microbiota and metabolites.

## 1. Introduction

Maternal nutrition plays a significant role in both farrowing performance and the postnatal growth of offspring. During gestation and lactation, sows experience endogenous and exogenous stress, leading to adverse reactions in their offspring, especially during the weaning period [1], such as low weight gain and diarrhea, which can result in high mortality and reduced growth rates in neonatal piglets [2]. It has been reported that modifying maternal nutrition or using functional additives can significantly improve offspring. For example, maternal soybean hull and cornstalk supplementation during gestation can increase the weaning weight of piglets [3], and sows supplied with enzymatically treated *Artemisia annua* L. show increased duodenal and ileal villus height in pre-weaning offspring [4]. Therefore, maternal nutrition is crucial in regulating the health and performance of weanling piglets.

Nowadays, the role of intestinal microbiota in the health of sows and their offspring is becoming increasingly prominent due to its close relationship with nutrient digestion, hormone secretion, and immune regulation [5,6]. Sows have a significant impact on the formation of the gut microbiome in their offspring. Almost all sources of microbiota for offspring are related to the mother, including maternal feces, maternal surface microorganisms, and the living environment. The transfer of microbes from sows to piglets may occur much earlier than previously thought, and the intestinal microbiota may be transmitted from the mother to the fetus during the gestation period [7]. Grzeskowiak et al. reported that sows fed high-fermentable fiber during gestation and lactation periods had increased microbial diversity and decreased susceptibility to *C. difficile* in their offspring [8]. One study has also shown that maternal probiotic supplementation decreases the levels of proinflammatory cytokines in the plasma and alters the microbiome composition in the colon of suckling piglets [9]. These studies suggest the possibility of improving the gut microbiome of piglets by using sow functional additives.

Many natural phytogenic additives are commonly used in livestock production diets as functional additives because they can scavenge free radicals, alleviate inflammatory responses, or modulate the immune-neuro system [10,11]. One such bioactive component is magnolol (MAG), a lignan isolated from *Magnolia officinalis*, which has been used in traditional Chinese medicine for a long time [12]. Numerous studies have indicated that MAG possesses significant antioxidant, anti-inflammatory, antibacterial, and anticancer properties [13,14]. The feeding efficacy of MAG has been studied. Xie et al. reported that MAG enhanced the growth performance and meat quality of broilers by improving their antioxidant and gut microbiota homeostasis [15]. Moreover, MAG also relieves oxidative stress and liver injury via the MAPK/mTOR/Nrf2 signaling pathway [16]. Furthermore, dietary supplementation with MAG improved hen performance, egg production, and egg quality [17]. In addition, MAG has been reported to alter placental morphology and enhance offspring growth in a pregnant mouse model [18].

In the present study, we hypothesized that dietary supplementation of sows with MAG throughout gestation and lactation might alter the oxidative status, diarrhea, intestinal barrier functions, and profile of the gut microbiome and metabolome of the offspring. These findings offer novel insights into the potential applications of MAG during pregnancy.

## 2. Materials and Methods

### 2.1. Animals and Diets

Fifty sows (Landrace × Yorkshire) up to parity 3 were randomly allocated into two dietary treatment groups (n = 25 per group) based on backfat thickness (BFT). The sows were fed either control diets (CONT) or a control diet supplemented with 100 g/t of MAG (ChengDu ConBon Biotech, Chengdu, China) throughout the entire gestation and lactation periods. Commercial control and pre-starter feeds were used. Their nutrient contents are shown in Appendix A. At the beginning of the experiment, sows were assigned to gestation crate (1.8 × 0.8 m) and were kept there until confirmed gestation (35 d). From there, the feed was offered two times a day at 0700 and 1500 until day 110 of gestation. Then, the sows were transferred to the farrowing house, placed in separate farrowing pens (2.4 × 3.8 m), and fed three times a day at 0600, 1300, and 1800. The sows allotted per treatment during farrowing were n = 22 for the CONT and n = 21 for MAG. During gestation and lactation periods, the sows had free access to water. The temperature in the farrowing house was automatically controlled, and parturitions were monitored to avoid interference during the farrowing process.

To standardize litter size, piglets were cross-fostered within their respective treatment groups of sows (CONT or MAG) to 12 piglets per litter within 1–2 days after birth. Sows with <6 piglets per litter were excluded, and a few of their babies were adopted within their respective treatment groups. Therefore, n = 15 was used in the CONT and MAG groups during lactation period. After weaning, the piglets were moved to a nursery house and divided into separate pens according to their mother groups and parity. Weanling piglets from 12 and 10 sows in second parity in the control and MAG groups, respectively, were assigned to the test nursery house. There, weanling piglets from one or two sows in the same group were placed in one nursery pen; a total of seven pens in the control group and six pens in the MAG group were used.

### 2.2. Data Recording

Sow BFT was measured on days 0, 30, 60, 90, and 110 of gestation and at weaning using a model WED-3000 V digital B-ultrasound device (Welld, Shenzhen, China) at the P2 position (left side of the midline at the last rib and 6.5 cm to the spine). The average daily feed intake (ADFI) of sows during gestation and lactation was recorded. The number of days from weaning to estrus (WOI) of sows was recorded.

Reproductive performance parameters included the litter size at farrowing, the total number of live piglets, and the individual piglet body weight (BW) at birth, cross-fostering, day 21 of lactation, and day 7 postweaning. From day 1 to 7 postweaning, the fecal scores of weanling piglets were recorded daily on a scale: 1 = normal solid stool, 2 = looser than normal stool, 3 = fluid stool, and 4 = watery stool [19].

### 2.3. Blood and Tissue Sample

One female piglet per pen (medium BW piglet per pen) and six piglets per group were euthanized 1 week after weaning. The blood samples were collected by jugular vein puncture, centrifuged at 3500 rpm for 15 min to obtain serum, and stored at −20 °C until analysis. The samples of duodenum, jejunum, ileum, and colon tissues (1 cm^2^) were collected immediately and stored at −80 °C until analysis. The Colonic luminal contents samples were also collected, quick-frozen in liquid N_2_ and stored at −80 °C until analysis.

### 2.4. H&E Staining

The fresh samples of duodenum, jejunum, and ileum were placed in 4% paraformaldehyde overnight and processed for H&E staining. Briefly, the tissues were sequentially treated with running water, a graded alcohol series, xylene and then embedded in paraffin. Sections of 7 μm were cut, deparaffinized, and rehydrated, followed by hematoxylin and eosin staining. The images were collected using biological microscope E100 (Nikon, Tokyo, Japan). The villus height and crypt depth of these pictures were calculated by ImageJ software (version 1.4.3.67).

### 2.5. Measurement of Serum Oxidative Status

The total antioxidant capacity (T-AOC), enzymatic activity of catalase (CAT) and superoxide dismutase (SOD), and the content of malondialdehyde (MDA) of serum were determined with kits from Nanjing Jiancheng Bioengineering Institute (Nanjing, China) according to the instructions. MDA was measured at 95 °C and 532 nm, while 570 nm for T-AOC, 540 nm for CAT, 450 nm for SOD at 37 °C by a spectrophotometric assay.

### 2.6. Measurement of Serum Diamine Oxidase (DAO)

According to the manufacturer’s protocol, DAO activity in serum was determined using a kit from the Nanjing Jiancheng Bioengineering Institute (Nanjing, China).

### 2.7. Quantitative Real-Time PCR

An RNAiso Plus kit (108-95-2, Takara, Shiga, Japan) was used for total RNA isolation. After purity and concentration, a measure of 1.5 μg total RNA isolated from colon tissue samples was used for cDNA synthesis using PrimeScript™ RT reagent Kit with gDNA Eraser (Perfect Real Time) (RR047A, Takara, Shiga, Japan). Quantitative real-time PCR of the expression of zona occludens-1 (ZO-1), claudin-1, occludin, and intestinal alkaline phosphatase (IAP) were analyzed in a LightCycle 96 Instrument (Roche, Mannheim, Germany). 18S ribosomal RNA (18S) was selected as the internal control. For statistical analysis, the relative expression levels of each gene were calculated using 2^−ΔΔCt^ method.

### 2.8. The 16S rRNA High-Throughput Sequencing

The samples of the colonic contents were collected for genomic DNA extraction using soil/fecal genomic DNA extraction kit (DP802, Tiangen Biotech, Beijing, China). After the extracted DNA was examined and purity determined, the hypervariable region V3-V4 of the bacterial 16S rRNA gene was amplified with primer pairs (338F: 5′-ACTCCTACGGGAGGCAGCA-3′ and 806R: 5′-GGACTACHVGGGTWTCTAAT-3′). The amplified products were purified with Omega DNA purification kit (Omega, Norcross, GA, USA) and quantified using Qsep-400 (BiOptic). Then, equal amounts of purified PCR products were paired-end sequenced (2 × 250) on an Illumina novaseq6000 (Beijing Biomarker Technologies, Beijing, China).

USEARCH (version 10.0) was used to assign qualified sequences with a similarity threshold exceeding 97% to one operational taxonomic unit (OTU). QIIME2 software was used to perform taxonomy annotation of the OTUs and alpha diversity of each sample. Beta diversity calculations were analyzed by principal coordinate analysis (PCoA) based on Bray–Curtis distances at the OTU level. Linear discriminant analysis (LDA) coupled with effect size (LEfSe) was conducted to evaluate the biomarkers for each group. The KEGG pathway analysis of the OTUs was inferred using Tax4Fun (version 1.0) [20]. The phenotypic prediction was performed by BugBase [21]. The online platform BMKCloud (https://www.biocloud.net accessed on 13 May 2023) was used to analyze the sequencing data.

### 2.9. LC-QTOF Analysis of Piglet Colonic Content Metabolites

Liquid chromatography–tandem quadrupole time-of-flight mass spectrometry (LC-QTOF) was used for non-targeted metabolomics analysis. The chromatographic separation was performed on Waters Acquity UPLC HSS T3 column (1.8 μm, 2.1 × 100 mm). The 0.1% formic acid aqueous solution and 0.1% formic acid acetonitrile consisted of a mobile phase with injection volume 1 μL. Waters Xevo G2-XS QTOF high-resolution mass spectrometer can be performed on both low-collision energy and high-collision energy at the same time. The parameters of the electrospray ionization source (ESI) ion source are as follows: Capillary voltage: 2000 V (positive ion mode) or −1500 V (negative ion mode); cone voltage: 30 V; ion source temperature: 150 °C; desolvent gas temperature 500 °C; backflush gas flow rate: 50 L/h; desolventizing gas flow rate: 800 L/h.

The identified metabolites were searched for classification and pathway information in KEGG, HMDB, and Lipid maps databases. The principal component analysis (PCA) and orthogonal partial least-squares–discriminant analysis (OPLS-DA) were used to judge and verify the reliability of differences between MAG and CONT groups. The variable importance in projection (VIP) value was calculated using multiple cross-validations. The metabolites identified investigated the significant *p*-value between groups using the student’s t-test. The thresholds for significantly different metabolites were FC > 1, *p*-value < 0.05, and VIP > 1. The difference metabolites of KEGG pathway enrichment significance were calculated using hypergeometric distribution test.

### 2.10. Statistical Analyses

Statistical analysis was carried out with GraphPad Prism software (6.0c) (GraphPad Software version 6.01). The one-way ANOVA was used for statistical analysis. Values are means ± SEM. The numbers of diarrhea piglets in the two groups were compared by χ^2^ test for statistical significance. The significance level was set at *p* < 0.05.

## 3. Results

### 3.1. Gestation and Lactation Performance of Sows

Table 1 shows that the addition of 0.01% MAG to the breeding sow diet did not have a significant effect on sow feed intake compared to the control group. During the gestation period, the BFT of the sows in both groups increased continuously as the pregnancy progressed. However, the BFT of sows in the MAG group tended to increase more at day 110 (*p* = 0.083) and significantly increased at weaning (*p* < 0.01) compared to the CONT group. Furthermore, the loss of BFT in lactating sows in the MAG group tended to be lower than in the CONT group (*p* = 0.068). The weaning-to-estrus interval (MOI) also tended to decrease after MAG supplementation in the MAG group (*p* = 0.074). These results suggest that dietary supplementation of MAG can improve the gestational and lactational performance of sows.

### 3.2. Reproductive and Litter Performance of Sows

Table 2 shows that maternal magnolol supplementation did not have a significant effect on sow birth weight, total piglets born, or piglets born alive compared to the control group. Additionally, there was no significant difference in litter performance between the CONT and MAG groups. However, the pre-weaning mortality rate was significantly lower in the MAG group (*p* < 0.05).

### 3.3. Organ Weight, Length, and Morphology of Intestine of Weanling Piglets

We have also shown that maternal MAG supplementation during gestation and lactation did not lead to significant changes in the weights of the heart, liver, spleen, lungs, or kidneys in Table 3. However, there were significant increases in the lengths of the duodenum (*p* = 0.05) and Jejunum + Ileum (*p* < 0.05) in the MAG group.

The histomorphological analysis further demonstrated that MAG improved the morphology of the jejunum (Figure 1A) by increasing the villus height (Figure 1B) and ameliorated the duodenum by increasing the villus height/crypt depth ratio (Figure 1D). These findings provide evidence that maternal MAG supplementation improved the intestinal morphology of weanling piglets.

### 3.4. Serum Antioxidant Capacity, Diarrhea, and Intestinal Barrier Function of Weanling Piglets

To assess the serum antioxidant capacity of weanling piglets, we measured MDA content, SOD activity, and T-AOC activity in the CONT and MAG groups. As illustrated in Figure 2A–D, the MAG group exhibited a significant decrease in MDA content (*p* < 0.05) and a significant increase in SOD activity (*p* < 0.05) and T-AOC activity (*p* < 0.05) compared to the control group, indicating that maternal MAG supplementation improves the antioxidant status of weanling piglets.

Diarrhea is among the most significant factors that affect the growth of weanling piglets [22]. Therefore, we determined the fecal scores of the piglets to evaluate the situation (Figure 2E). The findings revealed that MAG considerably decreased the fecal scores of piglets (*p* < 0.05). Additionally, we assessed the number of piglets with diarrhea in both groups from days 1–7 postweaning. The outcomes demonstrated that the total count of piglets suffering from diarrhea during the first week postweaning was significantly lower in the MAG group (Table 4; *p* < 0.001). Maternal MAG supplementation also lowered the diamine oxidase (DAO) activity (Figure 2F, *p* < 0.05) and raised the expression of Claudin-1, ZO-1, and IAP in the colon tissues of weanling piglets (Figure 2G, *p* < 0.05). These findings suggest that maternal MAG supplementation diminishes the redox imbalance and prevents diarrhea in weanling piglets.

### 3.5. Diversity of Colon Microbiota in Weanling Piglets

It has recently become evident that the colon plays a crucial role in controlling fecal water content [23]. Therefore, modifications in the composition of the colon microbiota have been associated with piglet diarrhea [24]. We collected luminal contents from the colon of weanling piglets in the MAG and CONT groups and conducted 16S rRNA sequencing. The rarefaction curve gradually flattened, indicating that the sequencing data covered nearly all the diversity in the samples (Figure 3A). The beta diversity between the two groups, presented by PCoA, showed that the MAG group (*p* > 0.05) did not form distinct clusters from the CONT group. Regarding the individual sample, the MAG group had no impact on the alpha diversity indices of ACE (Figure 3C), Chao1 (Figure 3D), Simpson (Figure 3E), and Shannon (Figure 3F) compared to those of the CONT group. These findings suggest that maternal MAG supplementation had no effect on the diversity of the colonic microbiota in weanling piglets.

### 3.6. Taxonomic Differences of Colon Microbiota in Weanling Piglets

The bar plots show that maternal MAG supplementation significantly alters the relative abundance of bacteria at different taxonomic levels in phyla (Figure 4A) and genera (Figure 4B). The results indicate differences in the species and abundance of dominant microorganisms between the CONT and MAG groups. LEfSe analysis reveals variations in bacterial taxa composition and specific microbiota with different abundances, as shown in Figure 4C,D. Cladogram analysis shows that *Paludibacteraceae* and *p-251-o5* in the CONT group, and *Clostridiaceae 1*, *Lachnospiraceae*, *Lachnospirales*, *Selenomoonadales*, *Negativicutes*, *T34*, *Mollicutes RF39*, and *Mollicutes* in the MAG group, are the main causes of differences in the structure of the colon microbiota in weanling piglets. We selected bacteria with LDA scores > 3.0 and *p* < 0.05 as biomarkers in the CONT and MAG groups. In the CONT group, the predominant bacteria belonged to *Eubacterium ruminantium group*, *Paludibacteraceae*, *bacterium enrichment culture clone DPF35*, *Ruminococcus* sp., *Anaerovorax*, and *p-251-o5* (LDA > 3, *p* < 0.05). Meanwhile, MAG treatment significantly increased the abundance of *Negativicutes*, *Selenomonadales*, *Tenericutes*, *Mollicutes*, *Mollicutes RF39*, *Clostridiaceae 1*, *Faecalibacterium prausnitzii*, *Faecalibacterium*, *Oscillospira*, *Prevotella* sp., *T34*, *Pseudoflavonifractor*, *Pseudoflavonifractor capillosus*, *Coprococcus*, *Lachnospirales*, *Lachnospiraceae*, and *Blautia obeum* (LDA > 3, *p* < 0.05). These results suggest that significant microbial variations exist between the CONT and MAG groups.

### 3.7. Predictive Analysis of Microbiota Function in Weanling Piglets

To predict the different functional compositions of the colon microbiome in the MAG and control groups, we used Tax4fun analysis. The significantly different Kyoto Encyclopedia of Genes and Genomes (KEGG) categories between the two groups were mainly related to “Environmental information processing” (Figure 5A). Additionally, the results of the differential categories showed that four categories accounted for more than 3% of the proportion of the colon microbiota, including “ABC transporters”, “Quorum sensing”, “Cellular community-prokaryotes”, and “Organismal Systems”.

We also conducted phenotype prediction using BugBase analysis (Figure 5B). Our results indicated that the relative abundance of aerobic bacteria was higher in the CONT group (*p* = 0.035), including *Campylobacteraceae* and *Lactobacillaceae*, while *Helicobacteraceae* dominated the MAG group. Correlation analysis showed that the marker bacteria had a higher correlation with intestinal length and integrity (Figure 5C). Specifically, *Faecalibacterium prausnitzii* was negatively correlated with DAO and fecal score (*p* < 0.05) and positively correlated with the length of the duodenum, Jejunum + Ileum (*p* < 0.05), Claudin-1 (*p* < 0.05), and IAP (*p* < 0.001). *Blautia obeum* was negatively correlated with DAO (*p* < 0.01) and positively correlated with the lengths of the Jejunum + Ileum (*p* < 0.001), ZO-1, and IAP (*p* < 0.05). *Anaerovorax* was negatively correlated with duodenum length (*p* < 0.01), Jejunum + Ileum (*p* < 0.05), Claudin-1 (*p* < 0.05), ZO-1 (*p* < 0.001), and IAP (*p* < 0.001) and positively correlated with DAO and fecal score (*p* < 0.05). Therefore, the identified marker bacteria were closely correlated with the indices of intestinal integrity.

### 3.8. Colonic Luminal Metabolomic Analysis of Weanling Piglets

We conducted untargeted metabolomic analysis to examine differences in colonic luminal metabolites between the CONT and MAG groups. PCA was employed to assess changes based on metabolomic data from the colonic content samples, and a clear separation was observed between the two groups (Figure 6A), indicating their distinctiveness. We further utilized OPLS-DA to identify and characterize metabolites (Figure 6B) and selected differential metabolites with a threshold score of VIP > 1, fold changes >2 or <0.5, and *p* < 0.05. Over 540 differential metabolites were identified, including 29 up-regulated and 511 down-regulated metabolites (Figure 6C,D, Appendix A). Functional analysis of these differentially expressed metabolites in KEGG revealed that MAG treatment significantly altered the pathways “Chlorocyclohexane and chlorobenzene degradation”, “Citrate cycle (TCA cycle)”, “Toluene degradation”, “Two-component system”, and “Vitamin B6 metabolism” in the colon of weanling piglets (Figure 6E).

Next, we evaluated the changes in lipid profiles between the MAG and CONT groups (Figure 7). The total lipid composition analysis revealed that the CONT group was enriched in phosphatidylinositol (PI), diradylglycerol (DG), and prenol lipid (PR), whereas phosphatidylcholine (PC), phosphatidylethanolamine (PE), and phosphatidyl serine (PS) were significantly increased in the MAG group (Figure 7A). The heatmap in Figure 7B depicts the lipids whose abundances differed significantly between the two groups. To further analyze the metabolomic data of PC, we looked at the unsaturated and saturated acyl chains among the two hydrophobic acyl chains. The results showed that the abundance of PCs with more than two unsaturated bonds increased in the MAG group (Figure 7C). Additionally, differential lipids belonging to PE, PS, and PI with relative abundances greater than 1% are shown in Figure 7D. These results suggest that MAG administration altered lipid profiles of weanling piglets, and PC was the most affected lipid.

### 3.9. Correlation Analysis

Correlation analysis of indices of intestinal length, antioxidant capacity, and intestinal integrity, differential metabolites, and gut microbiota are shown in Figure 8. In Figure 8A, PC(16:0/18:1(9Z)), PC(16:0/22:6(4Z,7Z,10Z,13Z,16Z,19Z)), PC(20:2(11Z,14Z)/P-18:1 (9Z)), PC(16:0/20:4(5Z,8Z,11Z,14Z)), PC(14:0/22:4(7Z,10Z,13Z,16Z)), PC(18:3(9Z,12Z,15Z)/18:1(11Z)), and PC(16:0/22:5(7Z,10Z,13Z,16Z,19Z)) were positively correlated with the length of duodenum and jejunum + ileum, ZO-1 and IAP expression (*p* < 0.05) and negatively correlated with DAO (*p* < 0.05). Meanwhile, PC(16:0/20:1(11Z)) and PC(14:1(9Z)/22:0) were positively correlated with DAO (*p* < 0.01) and fecal scores (*p* < 0.05) and negatively correlated with ZO-1 and IAP expression (*p* < 0.05). Moreover, PE(19:0/18:1(11Z)) and PE(18:1(11Z)/19:0) negatively correlated with SOD activity, ZO-1 and IAP expression (*p* < 0.05). As shown in Figure 8B, except *Ruminococcus* sp, gut microbiota was highly correlated with glycerophospholipid. Moreover, *Tenericutes*, *Mollicutes*, and *Mollicutes RF39* were correlated with PC(18:0/18:1(11Z)), PC(16:0/20:1(11Z)), PC(14:1(9Z)/22:0), PE(19:0/18:1(11Z)), PE(18:1(11Z)/19:0), PS(19:0/16:1(9Z)), PS(18:0/18:1(9Z)), and PI(18:0/18:1(9Z)) (*p* < 0.05). These results indicate that glycerophospholipids compositions were associated with antioxidant capacity, intestinal integrity, and gut microbiota.

## 4. Discussion

This study investigated the effects of maternal MAG supplementation on the redox status, diarrhea, intestinal barrier function, gut microbiome, and metabolome profiles of the offspring. To the best of our knowledge, such investigations have not been previously reported.

The slightly bitter taste of MAG did not affect the feeding behavior of sows at low doses (Table 1). Our findings indicate that MAG may increase the backfat thickness of sows during gestation and reduce losses during lactation, suggesting that it promotes glycolipid metabolism in sows. During pregnancy, glucose from the mother is the primary energy source for the embryo/fetus, necessitating an increase in glucose supply to meet maternal–fetal–placental requirements. This requirement continues to rise throughout the gestational period [25]. However, elevated blood glucose levels lead to higher insulin demands, eventually resulting in physiological insulin deficiency in sows. MAG has a significant antidiabetic effect [26]. In diabetic mice, Wang et al. reported that oral administration of MAG reduced the level of CYP2E1, a diabetes-inducing protein, in the liver [27]. Methylglyoxal (MGO), identified as a diabetes marker, can lead to excessive ROS, which can disrupt insulin secretion in pancreatic β cells when its concentration is increased [28,29]. MAG protects cells from excessive MGO-induced protein glycation by enhancing glyoxalase activity and reducing MGO levels [30]. Furthermore, several in vitro studies have suggested that MAG promotes glucose uptake in adipocytes and insulin secretion in pancreatic β cells [31,32]. These findings indicate that MAG regulates insulin levels and modifies glucose and lipid metabolism. Additionally, some studies have shown that insulin can promote re-estrus in sows [33], which might explain why MAG can shorten the weaning-to-estrus interval in sows. However, the mechanism behind the observed increase in backfat thickness in sows in this experiment remains unclear.

Our findings suggest that maternal supplementation with MAG could attenuate the redox imbalance of weanling piglets. MAG exhibits remarkable antioxidant capacity [34]; by enhancing the performance of sows, it can positively affect piglets through the placenta, breast milk, or feces, thereby enhancing antioxidant capacity and improving their resistance to stress. Additionally, we observed that maternal MAG supplementation improved intestinal length and reduced the occurrence of diarrhea in weaned piglets. These results indicate that MAG can significantly enhance the performance of weanling piglets, and its impact on gut microbiota could be a crucial aspect of this improvement.

Furthermore, we investigated the effect of maternal MAG supplementation on intestinal barrier function in weanling piglets. Reduced serum levels of DAO and increased expression of ZO-1, claudin-1, and IAP in colon tissue indicated that MAG could improve intestinal barrier function in weanling piglets. DAO is an intracellular enzyme in the intestinal epithelium that passes through the gut mucosa and into the peripheral blood when intestinal barrier function is impaired [35]. ZO-1 and claudin-1 are integral components of the intestinal barrier [36], and IAP regulates tight junction protein levels, maintaining intestinal barrier function [37]. IAP also plays a vital role in intestinal barrier function, and its activity and function are impacted by food components or metabolites [38]. Therefore, maternal MAG supplementation may improve intestinal barrier function in weanling piglets.

Our studies revealed that supplementing the maternal diet with MAG resulted in an increased relative abundance of beneficial bacteria such as *Faecalibacterium prausnitzii*, *Oscillospira*, and *Coprococcus* in the colon microbiota of weanling piglets, compared to the control group. These bacteria have been shown in recent studies to provide numerous health benefits. For example, *Faecalibacterium_prausnitzii* enhances the intestinal barrier function by acting on tight junctions and alleviates inflammation by producing anti-inflammatory metabolites. This bacterium is considered a next-generation probiotic because of its potential health-promoting effects [39,40,41]. Similarly, *Oscillospira* produces short-chain fatty acids, such as butyrate and propionate, which are essential for maintaining intestinal health. It also plays a crucial role in enhancing mucus production and maintaining the integrity of the intestinal epithelium and barrier function [42,43,44]. Moreover, Zhang et al. reported that the immune function of sows was positively correlated with *Coprococcus* [45]. Our study revealed a clear correlation between gut microbiota and intestinal integrity index. Thus, maternal MAG supplementation may modulate the gut microbiota composition in piglets, leading to improved intestinal integrity.

Maternal microorganisms have been found to play a crucial role in shaping the gut microbiota of offspring via transmission through milk, uterus, placenta, and feces. This transmission has been shown to impact the immune function, growth, and development of the offspring. Recent studies, such as the work of Rothschild et al., have highlighted the significant influence of environmental factors on the composition and structure of gut microbiota [46]. Our findings suggest that maternal MAG supplementation can alter the colonic environment of weaned piglets, as evidenced by the many differentially expressed microorganisms in the colon affected by maternal factors, which are involved in “Environmental information processing”. Furthermore, we observed the downregulation of a large number of differential metabolites in the colon luminal contents of the MAG treatment group, indicating that maternal MAG supplementation may exert regulatory effects on the metabolic activity of piglet colonic microorganisms. However, the underlying mechanism of this effect remains unclear and requires further investigation.

Our analysis of the differential metabolites using the KEGG pathway database revealed significant alterations in the following top pathways in response to maternal MAG supplementation: “Chlorocyclohexane and chlorobenzene degradation”, “Toluene degradation”, “Benzoate degradation”, and “Aminobenzoate degradation”. These changes resulted in a noticeable decrease in the levels of some intestinal toxic metabolites, such as p-cresol (FC = 0.19) and 4-Hydroxybenzoic acid (FC = 0.55) in the MAG group. These toxins, which are mainly derived from gut microbiota, have been shown to disrupt tight junctions, leading to intestinal barrier damage [47,48]. Thus, MAG supplementation might protect the intestinal barrier from microbial metabolic byproducts.

Except for toxic metabolites, our study also found that MAG administration altered lipid profiles in the colon. Studies have indicated that diarrhea was always accompanied by the modulation of glycerophospholipid classes or metabolism [49,50]. These alterations can be attributed to the important functions of glycerophospholipid on the intestinal barrier. Glycerophospholipids, which mainly include PE, PS, PC, and others, are the prominent membrane lipids [51,52]. PC is a key component of the intestinal mucus barrier, and exogenous PC supplementation improves intestinal barrier functions [53,54]. Moreover, the unsaturated acyl chain in PC can increase membrane fluidity [55]. However, unsaturated bonds are prone to oxidation. We found that maternal MAG supplementation significantly decreased the relative abundance of aerobic traits. Therefore, the high level of PCs with polyunsaturated acyl chains in this study prompted a stringent anaerobic environment in the colon of weanling piglets in the MAG group. The gut microbes are extremely sensitive to oxygen concentration [56], while a low oxygen environment promotes the proliferation and metabolism of anaerobic microorganisms. Our findings of beneficial bacteria such as *Faecalibacterium prausnitzii*, *Oscillospira*, and *Coprococcus* were anaerobic microorganisms, which highly positively correlated with PCs with polyunsaturated acyl chains. Moreover, Gao et al. reported that the levels of PE, PS, and PC were associated with the gut microbiota composition and Tight Junction protein expressions [57]. Our results have shown that maternal MAG supplementation increased the relative abundance of PC, PE, and PS, and decreased PI abundance. Thus, maternal MAG supplementation might alter glycerophospholipid classes to maintain gut health in piglets. Interestingly, *Tenericutes*, *Mollicutes*, and *Mollicutes RF39* were correlated with glycerophospholipids with monounsaturated bonds, but their internal connection is unclear.

## 5. Conclusions

Our results revealed that maternal MAG supplementation improved the redox status, attenuated diarrhea, and protected the intestinal barrier function of weanling piglets. These improvements may be attributed to alterations in the composition and structure of the gut microbiota and metabolomic profile. Our findings provide evidence to support the theory of maternal nutritional interventions to improve progeny gut health through maternal magnolol supplementation.

## Figures and Tables

**Figure 1 metabolites-13-00797-f001:**
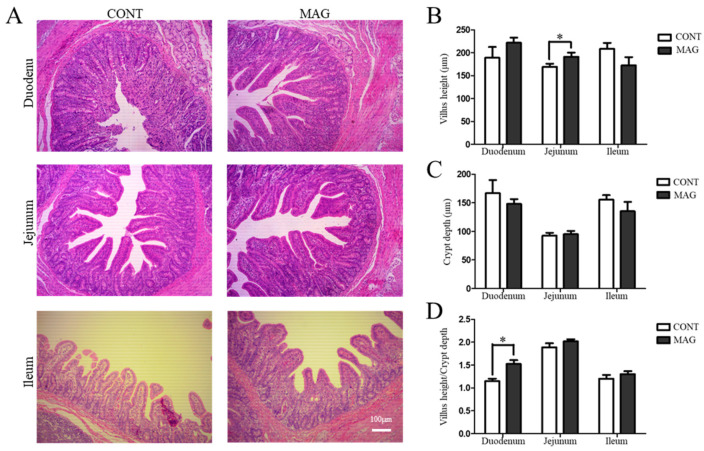
The effects of maternal magnolol supplementation on intestinal morphology in weanling piglets. (**A**) H&E-stained duodenum, jejunum, and ileum sections. (**B**) Villus height. (**C**) Crypt depth. (**D**) Villus height/crypt depth ratio. Scale bar 100 μm in A; n = 6 per group. * *p* < 0.05.

**Figure 2 metabolites-13-00797-f002:**
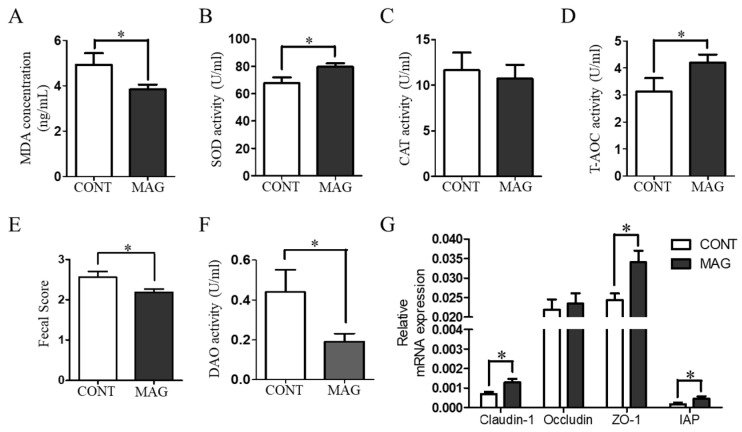
The effects of maternal magnolol supplementation on serum redox status, diarrhea, and intestinal barrier damage in weanling piglets. (**A**–**D**) Antioxidant indices, including MDA, SOD, CAT, and T-AOC. (**E**) Fecal scores of piglets. (**F**) Serum DAO activity. (**G**) The expression of Claudin-1, occludin, ZO-1, and IAP in colon tissue. n = 6 per group. * *p* < 0.05.

**Figure 3 metabolites-13-00797-f003:**
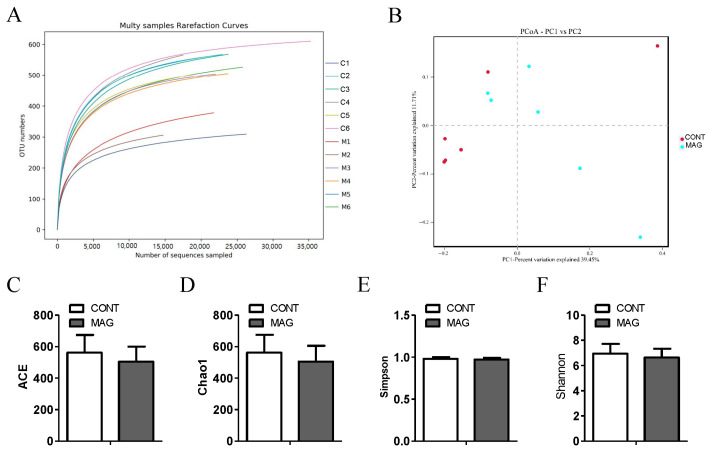
The effects of maternal magnolol supplementation on gut microbiota in weanling piglets. (**A**) Multisample rarefaction curves of OTU numbers detected from a randomly sampled sequence. (**B**) The beta diversity presented by PCoA plot based on unweighted UniFrac. (**C**–**F**) The alpha diversity indices of ACE, Chao1, Simpson, Shannon. n = 6 per group.

**Figure 4 metabolites-13-00797-f004:**
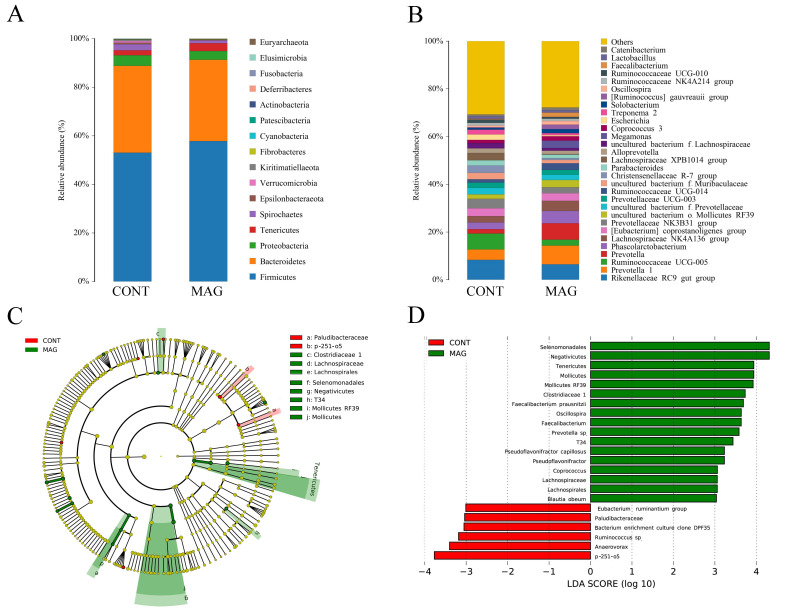
Comparison of gut microbiota composition in weanling piglets between CONT and MAG groups. The relative abundance of bacteria at different taxon levels in phylum (**A**) and genus (**B**), respectively. (**C**,**D**) Histogram and cladogram of taxonomic biomarkers identified in the colon microbiome data by LEfSe analysis. n = 6 per group. Differences are represented by the color of the most abundant class. Dot size is proportional to the abundance of the taxon.

**Figure 5 metabolites-13-00797-f005:**
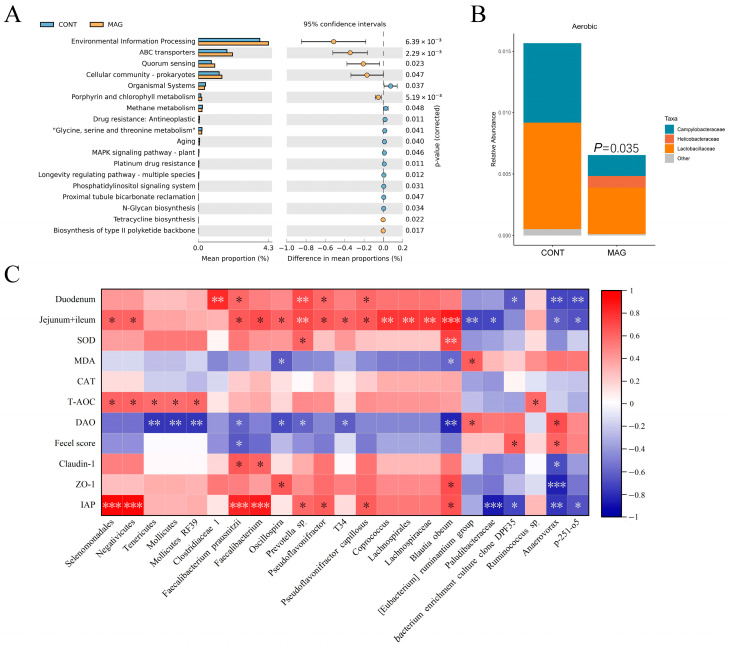
The bacterial functional potential prediction, phenotype prediction, and correlation analysis in weanling piglets. (**A**) Tax4Fun predictive analysis of the altered functional composition of gut microbiota under CONT and MAG groups. The KEGG pathways were analyzed by Tax4Fun and shown by STAMP. (**B**) BugBase analysis of microbiome phenotype prediction between CONT and MAG groups. (**C**) Heat map of Spearman’s rank correlation coefficient and significant test between marker bacteria and the indices of intestinal length, antioxidant capacity, and intestinal integrity. n = 6 per group. * *p* < 0.05, ** *p* < 0.01, and *** *p* < 0.001.

**Figure 6 metabolites-13-00797-f006:**
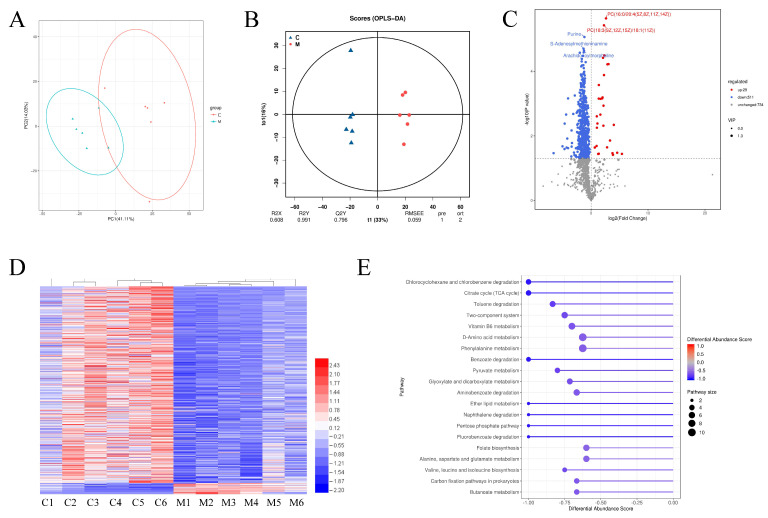
The differential metabolites and functional analysis of metabolome in colon of weanling piglets between CONT and MAG group. PCA (**A**) and OPLS-DA (**B**) score plots were used to compare the changes in the metabolic profile after magnolol treatment. (**C**) Volcano Plot of different metabolites. (**D**) Heat-map of the differential metabolites between CONT and MAG group. (**E**) KEGG pathway analysis of differential metabolites in weanling piglets by maternal magnolol treatment. n = 6 per group.

**Figure 7 metabolites-13-00797-f007:**
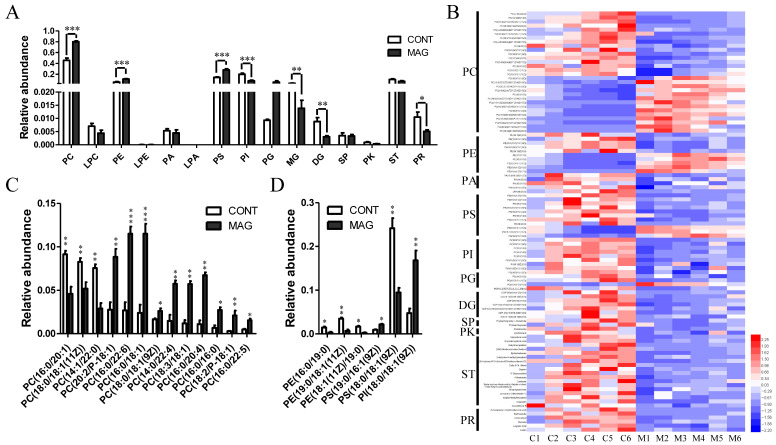
The analysis of the lipids whose abundance was significantly changed in MAG groups. (**A**) Relative abundance of the main lipid species in CONT and MAG groups. (**B**) Heat map of altered lipids between two groups. (**C**) Distributions of phosphatidylcholine (PC) with relative abundance > 1% in either of the two groups. (**D**) Distributions of phosphatidylethanolamine (PE), phosphatidylserine (PS), and phosphatidylinositol (PI) with relative abundance > 1% in either of the two groups. n = 6 per group. * *p* < 0.05, ** *p* < 0.01, and *** *p* < 0.001. LPC, lysoPC; LPE, lysoPE; PA, phosphatidic acid; LPA, lysoPA; PG, phosphatigylglycerol; MG, monoradylglycerol; SP, sphingolipid; PK, polyketide; ST, sterol lipid.

**Figure 8 metabolites-13-00797-f008:**
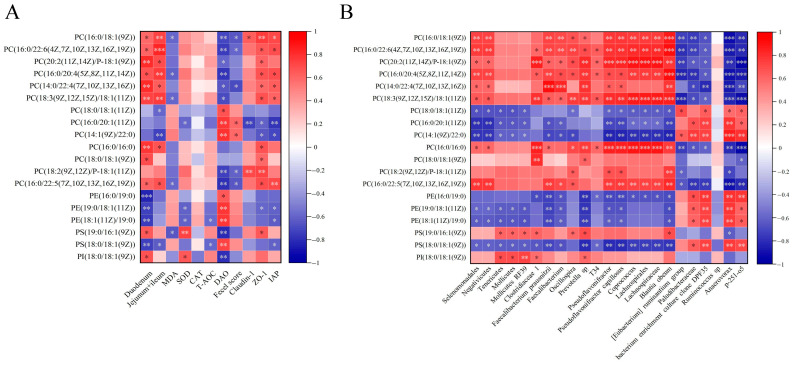
Correlation analysis among indices of intestinal length, antioxidant capacity, intestinal integrity, differential metabolites, and gut microbiota. (**A**) Correlation analysis between differential metabolites and indices of intestinal length, antioxidant capacity, and intestinal integrity. (**B**) Correlation analysis between differential metabolites and gut microbiota. n = 6 per group. * *p* < 0.05, ** *p* < 0.01, and *** *p* < 0.001.

**Table 1 metabolites-13-00797-t001:** Effects of dietary magnolol supplementation during gestation and lactation on performance of sows.

Item	Treatment		*p*-Value
	CONT	MAG	
No. of sows	25	25	-
Sow BFT, mm			
Breeding (day 0)	9.51 ± 2.27	9.2 ± 1.51	0.581
Gestation (day 110)	19.85 ± 3.26	21.14 ± 2.39	0.083
Weaning	16.61 ± 2.43	18.62 ± 2.19 *	0.0016
Loss lactation	3.57 ± 1.83	2.66 ± 1.63	0.068
Feed intake, kg			
Gestation ADFI (day 110)	2.86 ± 0.38	2.84 ± 0.42	0.921
Lactation ADFI (day 20)	6.05 ± 0.61	6.21 ± 0.57	0.847
WOI, days	5.50 ± 0.55	4.75 ± 0.71	0.074

* *p*-value < 0.05; the same as below.

**Table 2 metabolites-13-00797-t002:** Effects of dietary magnolol supplementation during gestation and lactation on reproductive sows and litter performances.

Item	Treatment		*p*-Value
	CONT	MAG	
Sow reproductive performance, n	22	21	-
Birth weight, kg	1.25 ± 0.43	1.28 ± 0.39	0.716
Total born piglets, n	9.23 ± 2.83	9.81 ± 3.06	0.219
Piglets born alive, n	9.00 ± 2.81	9.43 ± 3.22	0.534
Born alive piglet weight, kg	1.36 ± 0.33	1.31 ± 0.36	0.287
Litter performance, n	15	15	-
Litter size at cross-fostering (CF), n	12.1 ± 1.33	11.9 ± 0.53	0.919
Piglet CF weight, kg	1.41 ± 0.54	1.39 ± 0.32	0.663
Piglet lactation weight day 21, kg	6.03 ± 1.33	6.09 ± 1.30	0.812
Piglet postweaning weight day 7, kg	7.49 ± 1.16	7.25 ± 1.33	0.711
Preweaning mortality rate, %	2.09 ± 0.08	1.57 ± 0.06 *	0.047

* *p*-value < 0.05.

**Table 3 metabolites-13-00797-t003:** The weight of organs and the length of intestine in weanling piglets between CONT and MAG groups.

Item	Treatment		*p*-Value
	CONT	MAG	
No. of piglets	6	6	-
Body weight, kg	7.79 ± 0.26	7.85 ± 0.25	0.740
Heart, g	50.0 ± 7.07	47.5 ± 7.58	0.624
Liver, g	166.67 ± 18.07	174.32 ± 8.84	0.263
Spleen, g	18.33 ± 6.06	17.5 ± 4.18	0.792
Lung, g	120.83 ± 25.77	142.5 ± 27.88	0.109
Kidney, g	39.17 ± 5.85	43.33 ± 5.16	0.341
Duodenum, cm	27.25 ± 2.75	33.75 ± 4.95 *	0.050
Jejunum + Ileum, cm	723.5 ± 101.02	856 ± 60.42 *	0.041
Colon, cm	94.83 ± 12.86	90.17 ± 9.39	0.740

* *p*-value < 0.05.

**Table 4 metabolites-13-00797-t004:** Effects of maternal magnolol on the number of piglets with diarrhea from days 1–7 postweaning ^1^.

Day Postweaning	Number of Diarrhea Piglets/Total Piglets		χ^2^	*p*-Value
	CONT	MAG		
Day 1	6/140	2/111	0.514	0.473
Day 2	5/140	3/111	0.000	0.988
Day 3	8/140	1/111	2.652	0.103
Day 4	9/127	1/106	3.567	0.059
Day 5	7/127	1/106	2.205	0.138
Day 6	8/119	0/104 *	5.033	0.025
Day 7	9/119	1/104	3.815	0.051
Total in 1 week	52/912	9/753 *	22.158	0.000

^1^ Fecal score ≥ 3 termed diarrhea. * *p*-value < 0.05.

## Data Availability

The data presented in this study are available in the article.

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
