# Peer review of "Maternal Magnolol Supplementation during Pregnancy and Lactation Promotes Antioxidant Capacity, Improves Gut Health, and Alters Gut Microbiota and Metabolites of Weanling Piglets"

_metabolites, 2023, doi:10.3390/metabo13070797_

Round 1
Reviewer 1 Report
In this study submitted to Metabolites, the authors explored the effects of magnolol supplementation of sows during the gestation and lactation on the health of their offspring. The authors observed beneficial effects on sows and most importantly in piglets. They showed an improvement of redox balance, a reduction of diarrhea occurrence, an increase of gene expression of gut barrier markers. They performed 16S metagenomic analysis of colon microbiota as well as untargeted metabolomic analysis of colonic luminal metabolites, then correlation analyses between the numerous parameters studied.
Line 39 The reference 1 does not look accurate.
Line 66. Could you please introduce more the molecule magnolol. By instance, what is its chemical composition?
Line 79. The authors used a high number of sows (n=22 for the control group and n=22 for the MAG group). There were around 9 piglets born alive per sow. It means that the authors had 198 piglets for each group available for analysis. Only 6 piglets were analysed in table 3, 140 and 111 piglets in table 4, and 6 piglets in figures 3,4, 5, and 6. There is no indication of the sample size for figures 1, 2 and 7.
The sample size should always be detailed especially in the figure caption.
Why did the authors analysed so few piglets related to the available piglets and how did they choose the piglets to analyse?
The male and female piglets should be analysed independently. With so few samples, one expects that the authors analysed only one sex. Which one?
Line 95. The piglets were cross-fostered to 12 piglets per litter (it is confirmed in table 2), whereas the number of piglets born alive per sow was around 9. Does the increase of litter size from 9 to 12 had deleterious impact like undernutrition and contributed to preweaning mortality. Can the authors explain their choice of litter size?
Lines 100-109. Please make it clearer which analyses are done in sows and which ones are done in piglets.
Line 121. Did the authors write hydrochloric acid instead of hematoxylin?
Line 192. It is a trend. Please revise.
Line 204 Please add the p value.
Line 218. I am not sure that a morphology can be “enhanced”
Tables 2 and 4. Why was the preweaning mortality rate so high?
Figure 2. How was determined the fecal score?
Figure 2. I did not find which intestinal segment was analysed for redox status but I guess it is the colon because it is in the same figure than the barrier marker gene expression performed in colon. This should be detailed. Why did the authors assess these parameters in colon whereas their previous results showed more modifications in small intestine than in colon (table 3 and figure 1)?
Figure 4. Please increase the police. It was not possible to read and to review this part.
Same comment for figure 6.
Errors related to English language: Lines 25.29.111.115.
Typing error occludin line 137
Author Response
Response to Reviewer 1 Comments
Reviewer #1:
In this study submitted to Metabolites, the authors explored the effects of magnolol supplementation of sows during the gestation and lactation on the health of their offspring. The authors observed beneficial effects on sows and most importantly in piglets. They showed an improvement of redox balance, a reduction of diarrhea occurrence, an increase of gene expression of gut barrier markers. They performed 16S metagenomic analysis of colon microbiota as well as untargeted metabolomic analysis of colonic luminal metabolites, then correlation analyses between the numerous parameters studied.
Point 1: Line 39 The reference 1 does not look accurate.
Response 1: Thank you for pointing this out. A new reference “Maternal Nutrition During Late Gestation and Lactation: Association With Immunity and the Inflammatory Response in the Offspring” was used in line 38 in revised manuscript. This review discussed the effects of maternal nutrients on immunity and the inflammatory response in the offspring.
Point 2: Line 66. Could you please introduce more the molecule magnolol. By instance, what is its chemical composition?
Response 2: Thanks for your comment. A sentence of “a lignan isolated from Magnolia officinalis,” was added in line 63 on Introduction section in our revised manuscript.
Point 3: Line 79. The authors used a high number of sows (n=22 for the control group and n=22 for the MAG group). There were around 9 piglets born alive per sow. It means that the authors had 198 piglets for each group available for analysis. Only 6 piglets were analysed in table 3, 140 and 111 piglets in table 4, and 6 piglets in figures 3,4, 5, and 6. There is no indication of the sample size for figures 1, 2 and 7. The sample size should always be detailed especially in the figure caption.
Response 3: Thanks for your comment. The statements of the n number “n = 6 per group” were added in lines 234, 247, 279, 304, 334, 342, 374, and 399 on the figure captions in our revised manuscript.
Point 4: Why did the authors analysed so few piglets related to the available piglets and how did they choose the piglets to analyse? The male and female piglets should be analysed independently. With so few samples, one expects that the authors analysed only one sex. Which one?
Response 4: Thank you for pointing this out. After weaning, the piglets were moved to the nursery house and divided into separate pens according to their mother groups and parity. We have assigned the piglets from 12 and 10 sows in 2nd parity in control and MAG group respectively to the test nursery house. There, we have placed piglets from one or two sows in the same nursery pens, so 140 piglets in control group were placed into 7 pens, and 111 piglets in MAG group were placed in 6 pens.
Some studies focus on the impacts of maternal nutrition on offspring set the n number as 6 (Li et al. 2019; Zhou et al. 2022). In the comprehensive consideration of test needs and animal welfare, we have selected 1 female piglet per pen (medium BW piglet per pen) and 6 piglets per group to collect tissue samples.
Li Y, Liu H, Zhang L, Yang Y, Lin Y, Zhuo Y, Fang Z, Che L, Feng B, Xu S, Li J, Wu D. Maternal Dietary Fiber Composition during Gestation Induces Changes in Offspring Antioxidative Capacity, Inflammatory Response, and Gut Microbiota in a Sow Model. Int J Mol Sci. 2019, 21(1):31.
Zhou T, Cheng B, Gao L, Ren F, Guo G, Wassie T, Wu X. Maternal catalase supplementation regulates fatty acid metabolism and antioxidant ability of lactating sows and their offspring. Front Vet Sci. 2022, 9: 1014313.
The sentences of “Thus, the weanling piglets from 12 and 10 sows in 2nd parity in control and MAG group were assigned to the test nursery house. There, the weanling piglets from one or two sows in the same group were placed into a nursery pen, so 7 pens in control group and 6 pens in MAG group were used.” in lines 101-104, and “1 female piglet per pen (medium BW piglet per pen) and 6 piglets per group were” in line 117 were added on Materials and Methods section in our revised manuscript.
Point 5: Line 95. The piglets were cross-fostered to 12 piglets per litter (it is confirmed in table 2), whereas the number of piglets born alive per sow was around 9. Does the increase of litter size from 9 to 12 had deleterious impact like undernutrition and contributed to preweaning mortality. Can the authors explain their choice of litter size?
Response 5: We are very sorry for our negligence of sow reproductive performance. The sows with the number of piglets per litter < 6 was excluded, so there were 15 sows in control and MAG group respectively during lactation period. And the number of piglets per litter > 11.5 in control and MAG group (except the sows with low litter number). Many studies have set a litter size of 10 or 12 piglets each for the delivery-matched pairs during the cross-fostering operation (Mu et al. 2019; Yang et al. 2019). After consulting with the pig farm staff, we set a litter size of 12 piglets to retain more piglets for subsequent experiments.
Mu C, Bian G, Su Y, Zhu W. Differential Effects of Breed and Nursing on Early-Life Colonic Microbiota and Immune Status as Revealed in a Cross-Fostering Piglet Model. Appl Environ Microbiol. 2019, 18;85(9): e02510-18.
Yang Y, Hu CJ, Zhao X, Xiao K, Deng M, Zhang L, Qiu X, Deng J, Yin Y, Tan C. Dietary energy sources during late gestation and lactation of sows: effects on performance, glucolipid metabolism, oxidative status of sows, and their offspring1. J Anim Sci. 2019 Nov 4;97(11):4608-4618.
The sentences of “The sows allotted per treatment during farrowing was n = 22 for the CONT and n = 21 for MAG.” in lines 91-92, and “The sows with the number of piglets per litter < 6 was excluded and some of their babies were being adopted within their respective treatment groups. So, the n = 15 for the CONT and MAG groups during lactation period.” in lines 97-99 were added on Materials and Methods section in our revised manuscript. We have also made adjustments to the Table 2. The n number of sows was added in revised Table 2.
Point 6: Lines 100-109. Please make it clearer which analyses are done in sows and which ones are done in piglets.
Response 6: Thank you very much for your recommendation. Considering the Reviewer’s suggestion, we have disassembled the “2.2 Data recording” section into two paragraphs in our revised manuscript. The first paragraph was about the analyses are done in sows, and the second paragraph was about piglets.
Point 7: Line 121. Did the authors write hydrochloric acid instead of hematoxylin?
Response 7: We are very sorry for our incorrect writing. The statement of “hydrochloric acid and eosin staining” in line 121 has been changed to “hematoxylin and eosin staining” in lines 128-129 on Materials and Methods section in our revised manuscript.
Point 8: Line 192. It is a trend. Please revise.
Response 8: Thank you for pointing this out. The statement of “tended to” in line 203 has been added on Results section in our revised manuscript.
Point 9: Line 204 Please add the p value.
Response 9: Thanks for your comment. The statement of “(P < 0.05)” in line 216 has been added on Results section in our revised manuscript.
Point 10: Line 218. I am not sure that a morphology can be “enhanced”
Response 10: Thank you for pointing this out. As Reviewer suggested that the statement of “enhanced” in line 218 has been changed to “improved” in line 229 on Results section in our revised manuscript.
Point 11: Tables 2 and 4. Why was the preweaning mortality rate so high?
Response 11: We are very grateful for your crucial comment on this point. Temperature and sow nutrition have significant impacts on the early growth and healthy of piglets. In our test, the piglets were born at the end of November, with an outdoor temperature of around 0 degrees Celsius and a temperature of 22 degrees Celsius in the delivery room. Thus, the cold stress stimulation during breastfeeding and transfer to nursing houses can affect the health of piglets which leading to relatively high preweaning mortality rate.
Point 12: Figure 2. How was determined the fecal score?
Response 12: Thank you very much for your recommendation. Daily visual assessment monitoring of each piglet using a 4-point fecal scoring system: 1 = normal solid stool, 2 = looser than normal stool, 3 = fluid stool, and 4 = watery stool. We have detected the diarrhea situation of piglets one by one at 06:00 – 09:00 every day, observed fecal morphology and scoring, checked the redness and swelling of the anus of each piglet. After visual assessment has been done, we have also cleaned up the nursery pens every day.
Point 13: Figure 2. I did not find which intestinal segment was analysed for redox status but I guess it is the colon because it is in the same figure than the barrier marker gene expression performed in colon. This should be detailed. Why did the authors assess these parameters in colon whereas their previous results showed more modifications in small intestine than in colon (table 3 and figure 1)?
Response 13: We are very sorry for our incorrect writing. The statement of “2.5 Measurement of oxidative status” has been changed to “2.5 Measurement of serum oxidative status” in line 132 on Materials and Methods section in our revised manuscript. We have added “serum” in line 244 on Figure 2 caption in our revised manuscript.
Our results preliminarily demonstrate that the multifaceted effects of maternal magnolol supplementation on piglets. To our knowledge, the microbial content of the colon is more than 1000 times that of the small intestine, the roles and functions of colon microbiota deserve further research. Moreover, the colon plays a crucial role in controlling fecal water content. The modifications in the composition of the colon microbiota have been associated with piglet diarrhea. Thus, we have paid more attention to the colon. Thank you very much for your comments.
Point 14: Figure 4. Please increase the police. It was not possible to read and to review this part. Same comment for figure 6.
Response 14: Thanks for pointing this out. We have provided the figures with high resolution in our revised manuscript. We have also provided high resolution figures in PDF format attached to this text to avoid the impact of Word software on image quality.
Point 15: Comments on the Quality of English Language
Errors related to English language: Lines 25.29.111.115.
Typing error occludin line 137
Response 15: Thank you very much for your recommendation. The statement of “weanling piglets when sows” has been changed to “weanling piglets from sows” in line 24, “were correlation with” has been changed to “highly correlated with” in line 28 on Abstract section in our revised manuscript. The sentence of “and centrifuged at 3,500 rpm for 15 min to obtain serum and stored at –20 °C until analysis.” has moved to lines 119-120, and “, quick-frozen in liquid N2” in line 122 have been added on Materials and Methods section in our revised manuscript. The typing error of occludin has also been corrected in line 148 and Figure 2 in our revised manuscript.
Special thanks to you for your good comments.

Reviewer 2 Report
This is an interesting and exhaustive study that is focused on delineating the effects of maternal magnolol (MAG) supplementation on antioxidant capacity, gut health as well as gut microbiome and metabolome composition of weanling piglets. The results showed that maternal MAG supplementation improved the redox status of weanling piglets by decreasing MDA concentration, and increasing SOD, T-AOC activity. Maternal MAG supplementation also attenuated diarrhea concomitant with a decrease in serum DAO concentration as well as protected the intestinal barrier function of weanling piglets associated with an increase in the expression of colonic ZO-1, Claudin-1 and IAP. Microbiome analysis revealed that maternal MAG supplementation significantly increased the relative abundance of beneficial bacteria in colon of weanling piglets, including Faecalibacterium_prausnitzii and Oscillospira. Metabolome analysis showed an increase in glycerophospholipids such as phosphatidylcholine (PC), phosphatidylethanolamine (PE), and phosphatidylserine (PS) in the colon of piglets from MAG-fed dams. Moreover, these glycerophospholipids were found to be correlated with progeny gut health and the key beneficial bacteria in the piglets. The manuscript is easily understandable, straight-forward and the results are clearly presented. The abstract adequately summarizes the data and is concise. The findings provide evidence on the role of magnolol supplementation as a potential maternal nutritional intervention to improve progeny gut health. However, the authors should consider the following suggestions:
1. The manuscript lacks mechanistic data. It will be interesting to assess signaling mechanisms that are involved in mediating the antioxidant effects of maternal magnolol (MAG) supplementation.
2. Please add latest references (PMCID: PMC8866717 & PMCID: PMC9867015) that are associated with magnolol (MAG) supplementation studies.
3. Please carefully proof-read the manuscript for typo and grammatical errors.
Author Response
Response to Reviewer 2 Comments
Reviewer #2:
This is an interesting and exhaustive study that is focused on delineating the effects of maternal magnolol (MAG) supplementation on antioxidant capacity, gut health as well as gut microbiome and metabolome composition of weanling piglets. The results showed that maternal MAG supplementation improved the redox status of weanling piglets by decreasing MDA concentration, and increasing SOD, T-AOC activity. Maternal MAG supplementation also attenuated diarrhea concomitant with a decrease in serum DAO concentration as well as protected the intestinal barrier function of weanling piglets associated with an increase in the expression of colonic ZO-1, Claudin-1 and IAP. Microbiome analysis revealed that maternal MAG supplementation significantly increased the relative abundance of beneficial bacteria in colon of weanling piglets, including Faecalibacterium_prausnitzii and Oscillospira. Metabolome analysis showed an increase in glycerophospholipids such as phosphatidylcholine (PC), phosphatidylethanolamine (PE), and phosphatidylserine (PS) in the colon of piglets from MAG-fed dams. Moreover, these glycerophospholipids were found to be correlated with progeny gut health and the key beneficial bacteria in the piglets. The manuscript is easily understandable, straight-forward and the results are clearly presented. The abstract adequately summarizes the data and is concise. The findings provide evidence on the role of magnolol supplementation as a potential maternal nutritional intervention to improve progeny gut health. However, the authors should consider the following suggestions:
Point 1: The manuscript lacks mechanistic data. It will be interesting to assess signaling mechanisms that are involved in mediating the antioxidant effects of maternal magnolol (MAG) supplementation.
Response 1: We are very grateful for your crucial comment on this point. Magnolol has remarkable antioxidant and anti-inflammatory effects. The regulatory effect of magnolol combined with sow placental functions, or interaction of gut microbiota between mother and offspring would be the next topic of our research. Thank you very much for your guidance on our work.
Point 2: Please add latest references (PMCID: PMC8866717 & PMCID: PMC9867015) that are associated with magnolol (MAG) supplementation studies.
Response 2: Thank you very much for your recommendation. The ref. 15 and 16 were renewed and the sentences “Xie et al. reported that MAG can enhance the growth performance and meat quality of broilers by improving their antioxidant and gut microbiota homeostasis [15]. Moreover, magnolol also presented relieving effects on oxidative stress and liver injury via the MAPK/mTOR/Nrf2 signaling pathway [16].” was added in lines 67-70 on Introduction section in our revised manuscript.
Point 3: Please carefully proof-read the manuscript for typo and grammatical errors.
Response 3: Considering the Reviewer’s suggestion, we have applied for help from the language editor and fully utilized the grammar correction function provided by Word software. The sentences of “The control diet and pre-starter feed were the commercial feeds. Their nutrients contents were shown in Table S1.” in lines 85-86, and “a measure of 1.5μg of total RNAs isolated from colon tissue samples” in lines 145-146 have been corrected on Materials and Methods section in our revised manuscript. The “paired-end sequenced” has been changed to “paired-end sequencing” in line 160 on Materials and Methods section in our revised manuscript. Moreover, “Antioxidant indices, including MDA, SOD, CAT, and T-AOC.” in lines 245-246 on Results section, and “Faecalibacterium_prausnitzii has been found to enhance the intestinal barrier function by acting on tight junctions, alleviate inflammation by producing anti-inflammatory metabolites.” in lines 450-452 on Discussion section have been rewritten in our revised manuscript.
Thank you very much for your good comments.

Reviewer 3 Report
This study aimed to explore the effects of maternal magnolol (MAG) supplementation on antioxidant capacity, gut health as well as gut microbiome and metabolome composition of weanling piglets.
The experiment was properly designed and the statistics applied were appropriate.
I suggest small changes in order to make the manuscript easier to read:
1. It is not usual to say that the control group received 0 g of treatment per weight. So I suggest to change Abstract sentence “Fifty pregnant sows were randomly divided into two groups and supplemented with 0 and 100 g/t MAG during the gestation and lactation periods, and at 7 days postweaning the pups were euthanized” into “Fifty pregnant sows were randomly divided into two equally sized groups, the control group, and the group supplemented with 100 g/t MAG during the gestation and lactation periods, and at 7 days postweaning the pups were euthanized”.
2. There is no mention on DAO in Material and methods part of manuscript, please add it.
3. It could be useful to add an asterisk in tables second row (MAG), so significance could be easy detectable (where the p value is less then 0,05).
4. Please add explanation on the role of DAO in intestinal barrier function (Discussion part of manuscript, Lines 416-432). There are for the ZO-1, claudin-1, and IAP, but not for the DAO.
Author Response
Response to Reviewer 3 Comments
Reviewer #3:
This study aimed to explore the effects of maternal magnolol (MAG) supplementation on antioxidant capacity, gut health as well as gut microbiome and metabolome composition of weanling piglets. The experiment was properly designed and the statistics applied were appropriate.
I suggest small changes in order to make the manuscript easier to read:
Point 1: It is not usual to say that the control group received 0 g of treatment per weight. So I suggest to change Abstract sentence “Fifty pregnant sows were randomly divided into two groups and supplemented with 0 and 100 g/t MAG during the gestation and lactation periods, and at 7 days postweaning the pups were euthanized” into “Fifty pregnant sows were randomly divided into two equally sized groups, the control group, and the group supplemented with 100 g/t MAG during the gestation and lactation periods, and at 7 days postweaning the pups were euthanized”.
Response 1: We have made correction according to the Reviewer’s comments. The statement of “Fifty pregnant sows were randomly divided into two equally sized groups, the control group, and the group supplemented with 100 g/t MAG during the gestation and lactation periods, and at 7 days postweaning the pups were euthanized.” has been added in lines 16-19 on Abstract section in our revised manuscript. Thank you very much.
Point 2: There is no mention on DAO in Material and methods part of manuscript, please add it.
Response 2: Thanks for your recommendation. We have added “2.6 Measurement of serum diamine oxidase” on Material and methods section, and the statement of “The activity of diamine oxidase (DAO) in serum was determined with kit from Nanjing Jiancheng Bioengineering Institute (Nanjing, China), according to the manufacturer's protocol.” has also been added in lines 140-142 on Material and methods section in our revised manuscript.
Point 3: It could be useful to add an asterisk in tables second row (MAG), so significance could be easy detectable (where the p value is less than 0,05).
Response 3: Thank you for your suggestion. We have added the sentence of “*, presentation P-value is less than 0.05, the same as below.” in line 210 on Table 1 note. We have also added an asterisk in all tables second row (MAG) to indicate the significance.
Point 4: Please add explanation on the role of DAO in intestinal barrier function (Discussion part of manuscript, Lines 416-432). There are for the ZO-1, claudin-1, and IAP, but not for the DAO.
Response 4: Thanks for pointing this out. A new sentence has been used to explanation on the role of DAO as “DAO is an intracellular enzyme in the intestinal epithelium, which would pass through the gut mucosa and into the peripheral blood when intestinal barrier function impaired [35].” in lines 438-440 on Discussion section in our revised manuscript.
Once again, thank you very much for your comments and suggestions.

Round 2
Reviewer 1 Report
I checked the response and the revised version: all my comments have been addressed. The manuscript is suitable for publication.
-
Author Response
Response to Reviewer 1 Comments
Reviewer #1:
I checked the response and the revised version: all my comments have been addressed. The manuscript is suitable for publication.
Response: Thank you for your kind help and valuable comments on revising this manuscript. We have found a language editor to moderate editing of English language.
Abstract section
The sentences:
Lines 13-14 “The multifunctional bioactive component magnolol (MAG) has been shown to hold promise as a dietary supplement.” has been changed to “The multifunctional bioactive component magnolol (MAG) has shown promise as a dietary supplement.” in our revised manuscript.
The statements:
Line 19 “in colon of weanling piglets” has been changed to “of weanling piglet colons” in our revised manuscript.
Line 29 “highly correlated with” has been changed to “were highly correlated with” in our revised manuscript.
Line 31 ”which might be” has been changed to “possibly” in our revised manuscript.
The complete forms of MDA, SOD, T-AOC, DAO, ZO-1, and IAP were provided on Abstract section of our revised manuscript.
Introduction section
The statements:
Line 62 “due to their ability to” has been changed to “because they can” in our revised manuscript.
Line 68 “can enhence” has been changed to “enhanced” in our revised manuscript.
Lines 69-70 “magnolol also presented relieving effects on” has been changed to “MAG also relieves” in our revised manuscript.
Line 71 “has been shown to improve” has been changed to “improved” in our revised manuscript.
Line 75 “throughout the gestation and lactation periods may” has been changed to “throughout gestation and lactation might” in our revised manuscript.
The “magnolol” in line 61 and subsequent paragraphs has been replaced with “MAG”.
The “Additionally,” and “For instance,” in line 67 have been deleted on Introduction section of our revised manuscript.
Materials and Methods section
The sentences:
Line 85 “The control diet and pre-starter feed were the commercial feeds.” has been changed to “Commercial control and pre-starter feeds were used.” in our revised manuscript.
Lines 88-90 “Then, the sows were transferred to the farrowing house where they were placed in separate farrowing pens (2.4 × 3.8 m) and fed three times a day at 0600, 1300, and 1800.” has been changed to “Then, the sows were transferred to the farrowing house, placed in separate farrowing pens (2.4 × 3.8 m), and fed three times a day at 0600, 1300, and 1800.” in our revised manuscript. in our revised manuscript.
Lines 139-140 “The activity of diamine oxidase (DAO) in serum was determined with kit from Nanjing Jiancheng Bioengineering Institute (Nanjing, China), according to the manu-facturer's protocol.” has been changed to “According to the manufacturer's protocol, DAO activity in serum was determined using a kit from the Nanjing Jiancheng Bioengineering Institute (Nanjing, China).” in our revised manuscript.
The statements:
Line 82 “(n = 25)” has been changed to “(n = 25 per group)” in our revised manuscript.
Lines 97-98 “some of their babies were being adopted” has been changed to “a few of their babies were adopted” in our revised manuscript.
Line 98 “So, the n = 15 for” has been changed to “Therefore, the n = 15 was used in” in our revised manuscript.
Line 103 “into a” has been changed to “in one” in our revised manuscript.
Line 103 “so” has been changed to “a total of” in our revised manuscript.
Line 107 “BFT was measured by digital B-ultrasound (model WED-3000V, Welld, Shenzhen, China)” has been changed to “using a model WED-3000V digital B-ultrasound device (Welld, Shenzhen, China)” in our revised manuscript.
Line 114 “every day with a” has been changed to “daily on a” in our revised manuscript.
Line 118 “were selected to be euthanized at 1 week after weaning.” has been changed to “were euthanized 1 week after weaning.” in our revised manuscript.
Moreover, the words of “respectively” in line 102 and “(DAO)” in line 138 have been added on Materials and Methods section of our revised manuscript.
Discussion section
The sentences:
Lines 400-402 “The present study was conducted with the objective of investigating the effects of maternal MAG supplementation on redox status, diarrhea, intestinal barrier function, and the profile of the gut microbiome and metabolome in offspring.” has been changed to “This study investigated the effects of maternal MAG supplementation on the redox status, diarrhea, intestinal barrier function, gut microbiome, and metabolome profiles of the offspring.” in our revised manuscript.
Line 437-438 “, which would pass through the gut mucosa and into the peripheral blood when intestinal barrier function impaired” has been changed to “that passes through the gut mucosa and into the peripheral blood when intestinal barrier function is impaired” in our revised manuscript.
The statements:
Line 425 “of” has been changed to “with” in our revised manuscript.
Line 427 “it can also positively impact” has been changed to “it can positively affect” in our revised manuscript.
Line 448 “has been found to enhance” has been changed to “enhances” in our revised manuscript.
Line 451 “due to” has been changed to “because of” in our revised manuscript.
Line 451 “has been shown to produce” has been changed to “produces” in our revised manuscript.
Line 452 “like” has been changed to “, such as” in our revised manuscript.
Except for the modifications mentioned above, the modifications such as “the”, singular and plural are highlighted in red in our revised manuscript.
Thanks to your scientific and kind guidance.
